

# Biological effects of carbon nanotubes generated in forest wildfire ecosystems rich in resinous trees on native plants

Javier Lara-Romero[1,*], Jesús Campos-García[2,*], Nabanita Dasgupta-Schubert[3], Salomón Borjas-García[3], DK Tiwari[3], Francisco Paraguay-Delgado[4], Sergio Jiménez-Sandoval[5], Gabriel Alonso-Nuñez[6], Mariela Gómez-Romero[7], Roberto Lindig-Cisneros[7], Homero Reyes De la Cruz[2] and Javier A. Villegas[2]

[1] Facultad de Ingeniería Química, Universidad Michoacana de San Nicolás de Hidalgo, Morelia, Michoacán, México
[2] Instituto de Investigaciones Químico Biológicas, Universidad Michoacana de San Nicolás de Hidalgo, Morelia, Michoacán, México
[3] CONACYT-El Colegio de Michoacán/Ladipa, La Piedad, México
[4] Centro de Investigación en Materiales Avanzados S.C., Chihuahua, México
[5] Centro de Investigación y de Estudios Avanzados del IPN, Unidad Querétaro, Querétaro, México
[6] Centro de Nanociencias y Nanotecnología, Universidad Nacional Autónoma de México, Ensenada, Baja California, Mexico
[7] Instituto de Investigaciones en Ecosistemas y Sustentabilidad, Universidad Nacional Autónoma de México, Morelia, Michoacán, Mexico
* These authors contributed equally to this work.

Corresponding author
Javier A. Villegas,
vmoreno@umich.mx

## ABSTRACT

Carbon nanotubes (CNTs) have a broad range of applications and are generally considered human-engineered nanomaterials. However, carbon nanostructures have been found in ice cores and oil wells, suggesting that nature may provide appropriate conditions for CNT synthesis. During forest wildfires, materials such as turpentine and conifer tissues containing iron under high temperatures may create chemical conditions favorable for CNT generation, similar to those in synthetic methods. Here, we show evidence of naturally occurring multiwalled carbon nanotubes (MWCNTs) produced from *Pinus oocarpa* and *Pinus pseudostrobus,* following a forest wildfire. The MWCNTs showed an average of 10 walls, with internal diameters of ~2.5 nm and outer diameters of ~14.5 nm. To verify whether MWCNT generation during forest wildfires has a biological effect on some characteristic plant species of these ecosystems, germination and development of seedlings were conducted. Results show that the utilization of comparable synthetic MWCNTs increased seed germination rates and the development of *Lupinus elegans* and *Eysenhardtia polystachya*, two plants species found in the burned forest ecosystem. The finding provides evidence that supports the generation and possible ecological functions of MWCNTs in nature.

## INTRODUCTION

Carbon nanotubes (CNTs) have been the subject of extensive research in recent years because of their extraordinary properties and broad range of biotechnological applications. Although CNTs are commonly considered human-engineered nanomaterials, it has been generally accepted that nature may provide appropriate conditions for their synthesis. CNT occurrences have usually been sought in extreme environments (e.g., at high temperatures and pressures), where evidence has suggested their formation. For example, encapsulated CNTs have been found in the coal-petroleum mix of oil wells (*Velasco-Santos et al., 2003*) and in Greenland ice-core samples dated from the Neolithic Stone Age (10,000 years ago) (*Esquivel & Murr, 2004*); however, the source of these CNTs has not yet been identified. There have also been questions regarding the validity of these reports because of the lack of clear high-resolution transmission electron microscopy (HR-TEM) images, Raman analysis, or diffraction patterns (*Mackenzie et al., 2008*).

Previous studies have speculated that CNTs can form in volcanoes, based on the observation that Mount Etna's lava can catalyze the synthesis of multiwalled CNTs (MWCNTs) (*Su et al., 2008*; *Su & Chen, 2007*). However, no direct evidence of the formation of CNTs within volcanoes has been confirmed. Further, plant products such as turpentine, eucalyptus oil, neem oil, palm oil, and olive oil have been used as raw materials for chemical vapor deposition (CVD) in CNT synthesis (*Afre et al., 2005*; *Ghosh et al., 2007*; *Kumar, Tiwari & Srivastava, 2011*; *Suriani et al., 2009*). In addition, plant and fungal tissues containing transition metals have been used as natural catalyst precursors in the production of CNTs by CVD (*Zhao et al., 2011*).

Oleoresin extraction is commonly performed in the forestlands of Michoacán, México, where oleoresin is collected from the trunks of living pines, and turpentine is obtained from steam distillation. Alpha-pinene, which is used as a raw material for solvent production, is one of the most important components of turpentine documented as an effective compound from which high-quality and high-yield MWCNTs can be synthesized by CVD (*Lara-Romero et al., 2011*). Pine species such as *Pinus leiophylla*, *Pinus oocarpa*, *Pinus montezumae*, *Pinus pseudostrobus*, and *Pinus teocote* are considered the most important tree species for oleoresin extraction in the Mexican industry. The ecosystems in Michoacán, México associated with these species of conifers are prone to wildfires. During the drought season, wildfires can cause temperatures between 600 and 900 °C; this, coupled with the presence of turpentines (or alpha-pinene) and conifer tissues containing iron, provides conditions similar to those required for CNT formation in a process like CVD.

Moreover, MWCNTs have also been described as plant growth promoters, favoring seed germination and an increase in the fresh weight of tomato plants (*Khodakovskaya et al., 2012*; *Yang, Cao & Rui, 2017*). Recently, nanotechnology tools have developed CNTs for potential applications in agriculture, including crop protection, pollution control, waste management, pesticide detection, nanosensing, and as nanofertilizers (*De La Torre-Roche et al., 2012*; *Gogos, Knauer & Bucheli, 2012*; *Hong, Peralta-Videa & Gardea-Torresdey, 2013*; *Khodakovskaya et al., 2012*; *Yang, Cao & Rui, 2017*). Contrary to the beneficial applications of CNTs, negative effects of nanoparticles on edible plants have also been discussed
(*Miralles, Church & Harris, 2012*); thus, the known effects of MWCNTs on plants are still limited, as are the responses of the natural and agricultural ecosystems to human-engineered nanomaterials (*Yang, Cao & Rui, 2017*).

This report, as a first attempt to understand the roles of crystalline nanomaterials in plant ecosystems and to scarce evidence of naturally-formed MWCNTs in the biosphere. The main objective of this study was to provide evidence of spontaneously and naturally occurring MWCNTs from *Pinus* species following a forest wildfire event, and their possible effects on germination and development of species found in the burned forest ecosystem.

## MATERIALS & METHODS

### Sample collection from a pine forest

During the dry season (June 2012), samples of burned wood were randomly collected from mature trees of two different pine forest sites in Michoacán, west-central México, which had been recently affected by forest wildfires. The sites were 'Cerro Huashan, Nahuatzen' (19°38′35″N, 101°56′46″W; sampling *P. oocarpa* 2 weeks after fire extinguishment) and 'Cerro de la Cruz, Uruapan' (19°26′40″N, 102°2′56″W; sampling *P. pseudostrobus* and *P. montezumae* 8 weeks after fire extinguishment). At least 20 samples of each pinus species were collected from each forest wildfire site. Sampling was collected under the supervision of the Ministry of Environment and Natural Resources specifications (Nom-059-SEMARNAT-2010) and the conservation program for flora and fauna of the Pico de Tancítaro (APFFPT) from Michoacán, México; established by the Mexican decree law of august 19, 2009; and the Program for the Sustainable Management of Mountain Ecosystems Pico de Tancítaro, Michoacán, México (APFFPT-2009). Wood samples were ground and thoroughly mixed for further analyses.

### CNT analysis

Samples of burned wood from various types of pine trees were characterized by Raman spectroscopy, thermogravimetry (TGA), and high-resolution transmission electron microscopy (HR-TEM), at least 20 samples of each pinus species were analyzed. Raman spectroscopy was performed using a micro-Raman spectrometer (Labram System model Dilor) equipped with a 20 mW He-Ne laser emitting at 514 nm, a holographic notch filter (supertNotch-Plus; Kaiser Optical Systems, Inc., Ann Arbor, MI, USA), and a 256 × 1,024 pixel charge-coupled device (CCD) image recorder. All measurements were carried out at room temperature with no special sample preparation.

TGA was carried out using a microbalance (Chan D-200) (*Doudrick, Herckes & Westerhoff, 2012*), where 40–50 mg samples of burned wood from the different pine species collected after a natural fire and MWCNTs synthesized by spray pyrolysis of α-pinene/ferrocene were air-heated between 25 and 700 °C at a rate of 5 °C/min, to obtain TGA combustion curves of the samples.

HR-TEM micrographs were obtained from a Philips CM-200 analytical TEM operating at 200 kV. Specimens for HR-TEM analysis were prepared by dispersing the samples in acetone through sonication for 2 min and air-drying a drop of the suspension on a perforated, carbon-coated Cu° grid.

## Seed germination and plant pot-growing using synthetic MWCNTs

Seeds of *Lupinus elegans* and *Eysenhardtia polystachya*, collected from the pine forest of Michoacán, México, were sterilized with 95% sulfuric acid for 20 min and by soaking in 1% sodium hypochlorite (NaOCl) for 3 min, respectively; both were then rinsed with sterile distilled water. The seeds of each species were divided into six separate sets of 100 seeds and incubated in a suspension of 0 (Control), 10, 20, 30, 40, or 50 µg/mL MWCNTs (Sigma-Aldrich, St. Louis, MO, USA; Cat. No. 698849; CVD-produced synthetic multiwalled CNTs, OD = 6.0–13.0 nm, ID = 2.0–6.0 nm length = 2.5–20 µm, average wall thickness 7–13 graphene layers, >98% purity) for 10 min. The MWCNTs were dispersed in water by a three-step acid treatment. The first step consists on ultrasonic mixing the MWCNTs with concentrated HCl for 4 h, after refluxing MWCNTs in nitric acid for 8 h at 80 °C, and finally refluxing sample in a 1:1 mixture of sulfuric and nitric acids for 4 h at 80 °C. Each seed set was then placed on moistened filter papers in five Petri plates (20–30 seeds per plate) and randomly distributed in a germination chamber. Germination was evaluated after 10 days of incubation at 26 °C with a 12:12 light/dark cycles.

Pot-growing tests examined samples from all six treatments, each with 12 replicates (72 plants in total for each seed type). Previously sterilized seeds (as described above) were directly planted in 5-cm-diameter polyethylene containers filled with 375 mL of the growth medium (Creci-root) provided by a local nursery. These containers were then divided into six separate sets and seeds were treated directly with 1.0 mL of a suspension of either 0 (Control), 10, 20, 30, 40, or 50 µg/mL MWCNTs, then covered with ~1.0 cm of plant growth substrate. The containers were arranged at random in trays and watered on alternate days for five weeks. At the end of the 5-week period, the plants were harvested, and biometric variables (leaf area and, fresh and dry weights of shoots and roots) were recorded. Data were statistically analyzed using Graph Pad software with an analysis of variance (one-way ANOVA), and mean were compared using Tukey's post hoc tests at a significance level of $p < 0.05$.

## RESULTS

### Identification and characterization of CNTs in the burned wood of resinous forests, after wildfire

Burned wood samples were collected after an intense wildfire in a resinous pine forest in the Michoacán state of Mexico (June 2012). This forest mainly comprised *P. oocarpa*, *P. pseudostrobus*, and *P. montezumae*. Samples of the carbonized trees of these species were first analyzed by Raman spectroscopy. The Raman spectra of three different burned wood samples indicate that *P. oocarpa* and *P. pseudostrobus* samples show characteristic bands for CNTs, *i.e.,* the *D* and *G* bands (Fig. 1A). The *D* band was observed at approximately 1,370 cm$^{-1}$, and the *G* band, also known as the tangential band, was observed at approximately 1,600 cm$^{-1}$, which arises from the $E_{2g}$ mode of the graphite plane and confirms the presence of sp$^2$ electronic hybridization in the carbon bond network. Unexpectedly, the 2*D* (*G'*) band, which is associated with the source or metal load and temperature during synthesis, was not found in the Raman spectra. Moreover, no CNT signals were detected in the samples of burned tree bark obtained from *P. montezumae* (Fig. 1A).

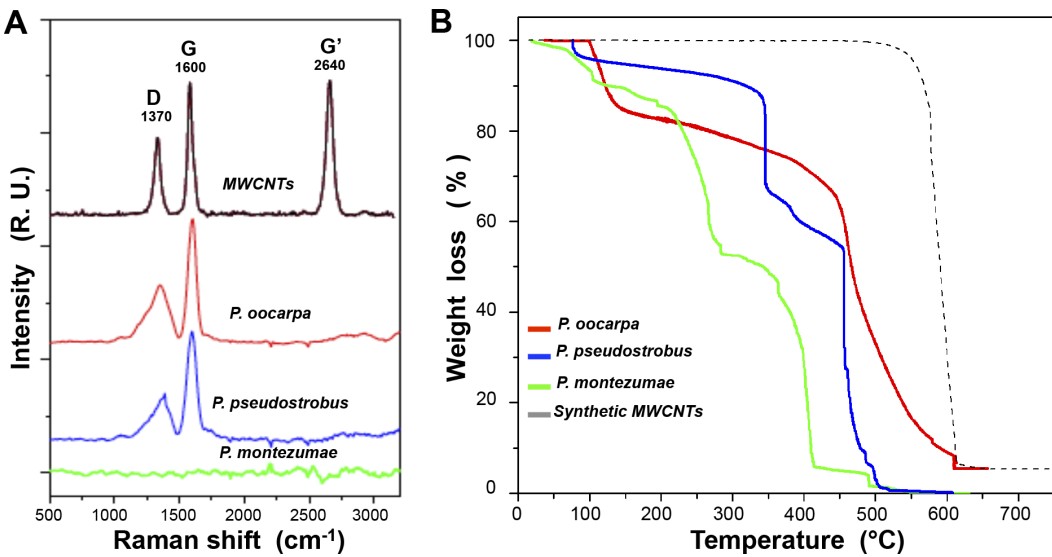

**Figure 1** **Analysis of the burned wood samples of *Pinus* species collected after a forest wildfire event.**
(A) Raman scattering spectra (He-Ne laser emitting at 514 nm) of the *Pinus* burned wood samples of:
(black) MWCNTs produced by chemical vapor deposition method using alpha-pinene/ferrocene as raw
material, (red) *P. oocarpa*, (blue) *P. pseudostrobus*, and (green) *P. montezumae*. The characteristic bands
of CNTs, *i.e.*, the D band (1,370 cm$^{-1}$), G band (1,600 cm$^{-1}$), and G' band (2,640 cm$^{-1}$) are shown.
(B) Thermogravimetric analysis (TGA) of the burned wood samples from *P. oocarpa*, *P. pseudostrobus*,
*P. montezumae*, and synthetic MWCNTs (pyrolyzed at 610 °C).

Thermogravimetric analysis (TGA) was used to determinate the amount of MWCNTs
in the burned wood of *P. oocarpa*, *P. pseudostrobus*, and *P. montezumae* (Fig. 1B). Weight
losses up to ∼150 °C correspond to the release of water contained in the samples, whereas
weight losses in the range of 200–300 °C and 300–400 °C are attributed to the degradation
of hemicellulose and cellulose, respectively. Weight losses in the range of 370–550 °C are
attributed to the ligneous components such as biochar (*Esquivel & Murr, 2004*; *Mackenzie
et al., 2008*; *Velasco-Santos et al., 2003*). Relevantly, the weight loss detected at 610 °C in
the *P. oocarpa* samples, which coincides with that in a synthetic-origin MWCNTs sample,
corresponds to CNT combustion. Thus, according to the TGA analysis, *P. montezumae*
contains approximately 10% (w/w) moisture, 38% (w/w) hemicellulose, 46% (w/w)
cellulose, and 4% (w/w) ligneous species; *P. pseudostrobus* is composed of approximately
5% (w/w) moisture, 7% (w/w) hemicellulose, 22% (w/w) cellulose, and 66% (w/w) of
ligneous components; and *P. oocarpa* is composed of approximately 14% (w/w) moisture,
18% (w/w) hemicellulose and cellulose, and 60% (w/w) ligneous components. Relevantly,
the TGA plot indicates that the burned wood samples of *P. oocarpa* contained ∼2.8% (w/w)
of CNTs and *P. pseudostrobus* less than 0.1% (w/w), and the remaining weight of ∼5–10%
(w/w) is attributed to metals and elements.

HR-TEM images and fast Fourier transforms (FFTs) of the *P. oocarpa* samples clearly
indicated the presence of CNTs. HR-TEM images and their corresponding FFTs show
diffraction patterns characteristic of graphitic crystalline carbon (Figs. 2A–2C). HR-
TEM data obtained from *P. oocarpa* samples revealed the presence of highly crystalline
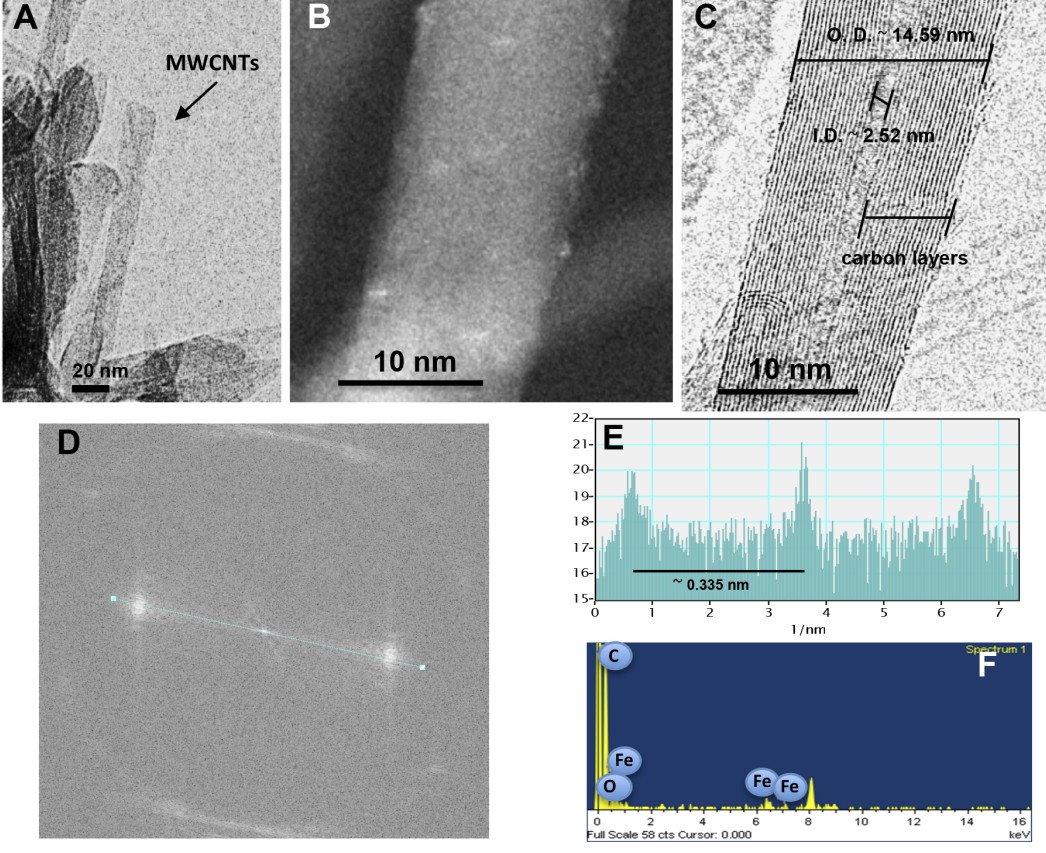

**Figure 2** **Identification of MWCNTs in the burned wood samples of *Pinus oocarpa* collected after a forest wildfire event.** (A–C) HR-TEM images of the burned wood samples at different magnifications, (D) FFT image, (E) analysis of the FFT image, and (F) EDS analysis. Representative images are shown.

MWCNTs, consisting of 10 walls with inner and outer diameters of ∼2.52 nm and ∼12–15 nm, respectively (Fig. 2C). The FFT image displayed one pair of sharp spots, and a line scan along those spots confirmed the presence of sharp spots corresponding to highly ordered carbon (narrow spots). The estimated plane-to-plane distance between the walls is 0.335 nm, which is in agreement with the nominal distance between the planes in crystalline CNTs (Figs. 2D–2E). The bright spots in the dark-field HR-TEM images indicate the presence of metals on the carbon tubes, and the corresponding energy-dispersive X-ray spectroscopy (EDS) analysis confirmed the presence of iron (Fig. 2F), suggesting that this iron could have acted as a catalyst during CNT formation.

For the burned wood samples of *P. pseudostrobus*, HR-TEM images and the corresponding FFT data show preferential formation of coil-shaped nanoparticles consisting of curved crystalline multiwalled carbon layers (Fig. 3A). The FFT images of these MWCNTs reveal one pair of sharp spots and a line scan along those spots confirmed the presence of highly ordered carbon. The estimated distance between the lattice fringes of the carbon walls is 0.335 nm, which is in agreement with the nominal distance between the planes of graphite (Figs. 3B–3C). The corresponding EDS analysis reveals the presence of several elements such as calcium, potassium, and phosphorous, but no evidence of

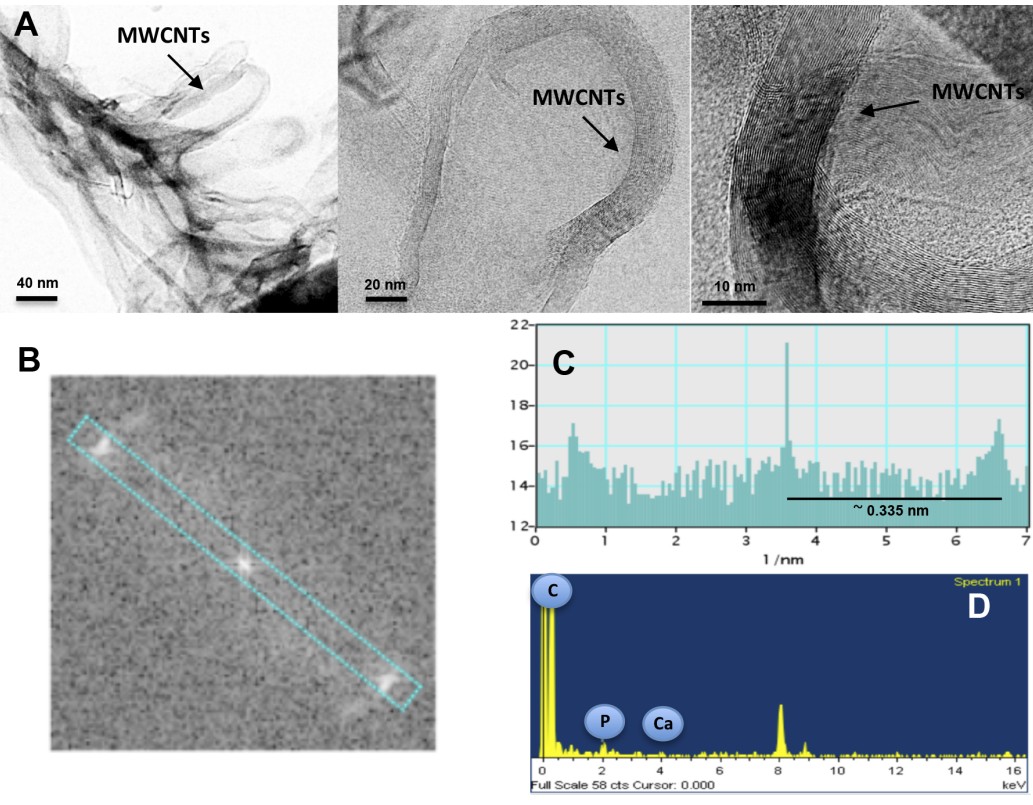

**Figure 3** **Identification of MWCNTs in the burned wood samples of *Pinus pseudostrobus* collected after a forest wildfire event.** (A) HR-TEM images of the burned wood samples at several magnifications, (B) FFT image, (C) FFT analysis, and (D) EDS analysis. Representative images are shown.

the presence of iron or other transition metals is found (Fig. 3D). Unexpectedly, no evidence of CNT structures was found in *P. montezumae* samples; however, amorphous carbon structures were abundant (Figs. 4A–4B). The FFT spectrum displayed diffuse spots, characteristic of amorphous carbon, and EDS analysis confirmed the presence of iron, calcium, and phosphorous (Figs. 4C–4E). These findings provided evidence of naturally occurring MWCNTs from *Pinus* species after forest wildfire events.

## Synthetic multiwalled CNTs increase seed germination in plants growing in resinous Pinus forests

The MWCNTs found in burned *P. oocarpa* and *P. pseudostrobus* wood samples had ∼10 layers, with an inner diameter of ∼2.52 nm and an outer diameter of ∼14.59 nm. To investigate if MWCNTs with structural features similar to those found in the natural samples could have a biological effect over some plants species characteristic of these ecosystems (*L. elegans* and *E. polystachya*), we conducted a germination and development of seedlings test. This assay was based on previous studies on the positive or negative effects on plant germination and the development of seedlings grown by MWCNT treatment (*Hong, Peralta-Videa & Gardea-Torresdey, 2013*; *Khodakovskaya et al., 2012*; *Miralles, Church & Harris, 2012*). We supplemented the seed germination and early seedling growth with 10–50 µg/mL of the synthetic MWCNTs with structural features similar to those found in

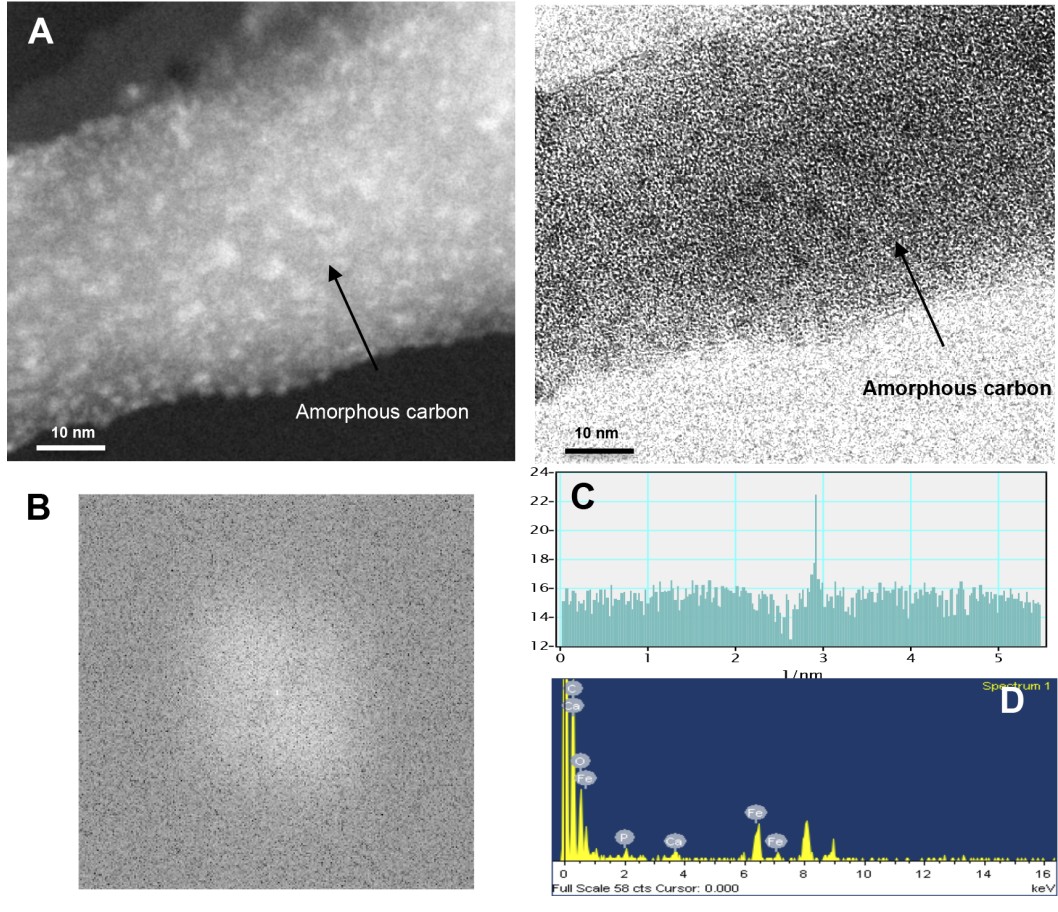

**Figure 4** **Identification of the carbon structures in the burned wood samples of *Pinus montezumae* collected after a forest wildfire event.** (A) HR-TEM images of the burned wood samples at the same magnifications, (B) FFT image, (C) FFT analysis, and (D) EDS analysis. Representative images are shown.

the *P. oocarpa* and *P. pseudostrobus* wood samples of burned forest (average wall thickness ∼7–13 layers; inner diameter of ∼2–6 nm; outer diameter of ∼12–20 nm; length of 2.5–20 µm).

Seed germination results showed that the addition of MWCNTs increased the number of germinated seeds and significantly shortened the germination period (Fig. 5A). Seeds of *L. elegans* and *E. polystachya* treated with MWCNTs exhibited increased germination rates compared to untreated seeds. For *L. elegans*, a prolific plant in this forest, seed germination rates were 62.5% higher after the addition of 30 µg/mL of the synthetic MWCNTs compared to those of untreated plants. Moreover, *E. polystachya* seeds treated with MWCNTs reached germination rates 40% higher than those of the untreated seeds (Figs. 5A–5C).

We further investigated the effects of MWCNTs on the growth and development of *L. elegans* and *E. polystachya* seedlings by growing them in a medium supplemented with different concentrations of the synthetic nanoparticles and measuring the yield of variables such as fresh and dry plant biomass, number of lateral roots, and foliar area (Figs. 6A–6B). *L. elegans* plants germinated and grown in the MWCNT dose range of 10–50 µg/L exhibited

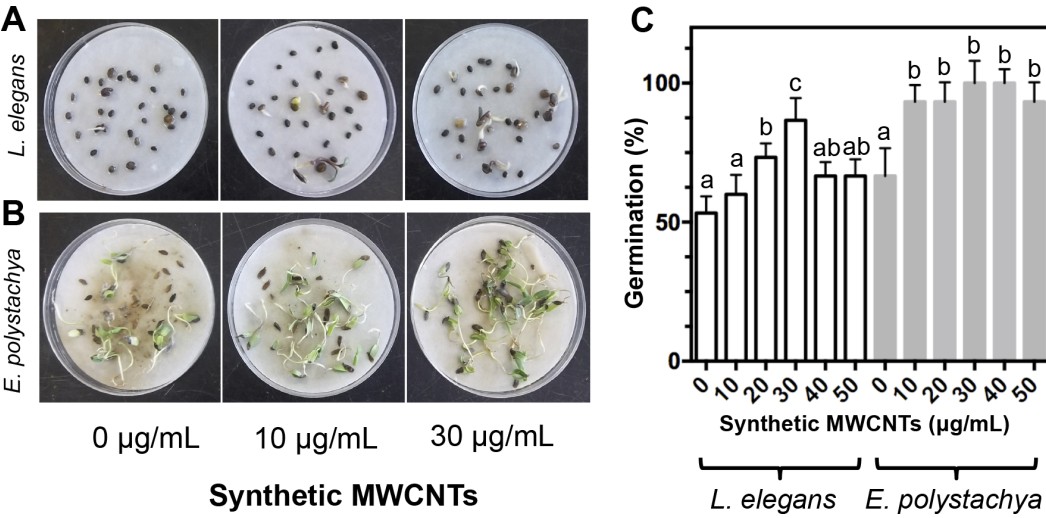

**Figure 5 Effect of synthetic MWCNTs on the seed germination rate of *Lupinus elegans* and *Eysenhard­tia polystachya*.** Seed germination of the native plants from the *Pinus* forest was evaluated after 10 days with MWCNTs treatment and recorded after 5-week of cultivation. (A) *L. elegans* seed germination, (B) *E. polystachya* seed germination, (C) quantitative data (A) and (B) assays. Bars represent mean ± standard error of three independent experiments, $n = 30$ each. One-way analysis of variance (ANOVA) was carried out with Tukey's *post hoc* test; statistical significance ($P < 0.05$) between treatments with respect to con­trol is indicated with different lowercase letters.

a significant amount of vegetative biomass at 30 µg/L and a decrease at 50 µg/L of MWCNTs (Fig. 6A). Significant increases in the fresh weight of the shoot and root, dry weight of the shoot, number of lateral roots, and foliar area (90.23%, 132.59%, 84.51%, 91.05%, and 93.72%, respectively) were observed in treated plants, compared to the untreated plants (Figs. 6C–6H). Results also suggest that the plant growth stimulation correlates with the increment in the shoot and root dry weights; when plants were treated with 30 µg/L of MWCNTs these variables reached a maximum of 45.2% and 120.46%, respectively, compared to those of the untreated plants (Figs. 6E–6F). *E. polystachya* plants grown in media supplemented with increasing doses of MWCNTs (10–50 µg/L) showed a large vegetative biomass at 50 µg/L and no negative effects of these nanotubes were recorded at any dosage level tested (Fig. 6B). The maximum increases in the shoot and root fresh and dry weight, number of lateral roots, and foliar area (87.8%, 302.78%, 148%, 114.54%, 313.66%, and 150.39%, respectively) were observed in treated plants, compared to the untreated ones (Figs. 6C–6H). These results show that synthetic MWCNTs increase seed germination and plant growth in two plant species growing in the studied resinous *Pinus* forests ecosystem.

## DISCUSSION

Our observations clearly support the hypothesis that MWCNTs can be formed spontaneously in nature and are capable of self-assemble without human interference. Although the process was not directly studied, the formation of MWCNTs during a resinous forest wildfire could be the consequence of a synthetic CVD-like mechanism.

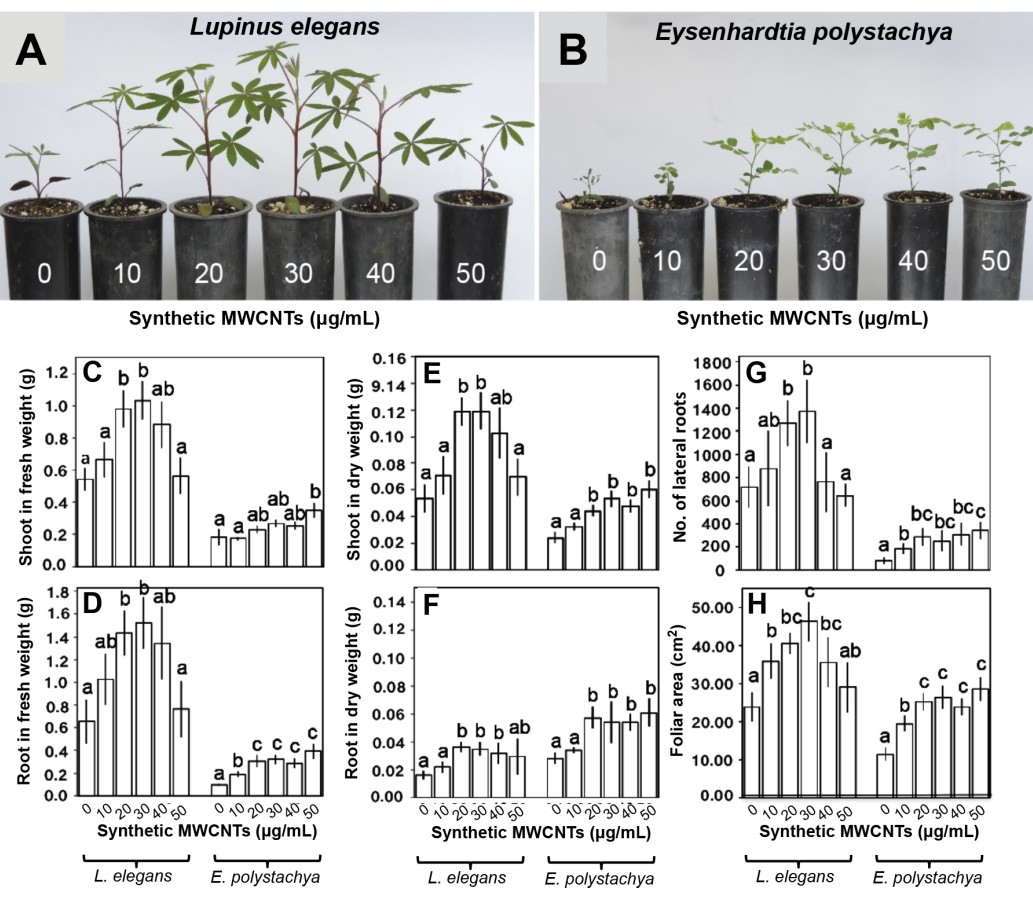

**Figure 6  Effect of synthetic MWCNTs on the plant growth rate of *Lupinus elegans* and *Eysenhardtia polystachya*.** After seed germination of the native plants from the *Pinus* forest (as described above), they were planted in 5–cm-diameter polyethylene containers filled with growth medium (Creci-root). These containers were then divided into six separate sets and the seedlings were treated directly with 1.0 mL of the suspension consisting of either 0 (control), 10, 20, 30, 40, or 50 µg/mL of synthetic MWCNTs. At the end of the 5-week period, the plants were harvested, and biometric variables were recorded. (A) *L. elegans* plant growth, (B) *E. polystachya* plant growth, (C–H) determination of the growth variables from (A) and (B) assays: (C) shoot in fresh weight, (D) root in fresh weight, (E) shoot in dry weight, (F) root in dry weight, (G), lateral roots number, and (H) foliar area. Bars represent mean ± standard error of three independent assays, $n = 72$. One-way analysis of variance (ANOVA) was carried out with Tukey's *post hoc* test; statistical significance ($P < 0.05$) between treatments with respect to control is indicated with different lowercase letters.

Production of MWCNTs by CVD requires the presence of volatile carbon compounds, which may act as precursors, in the gaseous state. *P. oocarpa* is a species rich in turpentine, and its oleoresin is a mixture of highly volatile monoterpenes, including $\alpha$- and $\beta$-pinenes, which have been identified as highly effective MWCNT precursors capable of providing a high yield (*Lara-Romero et al., 2011*). According to previous studies, coil-shaped crystalline nanoparticles cannot be synthesized by processes other than CVD (*Fejes & Hernádi, 2010*; *Mhlanga et al., 2011*). Therefore, the above hypothesis is also supported by the detection of coil-shaped crystalline carbon nanoparticles in the HR-TEM images of the burned *P. oocarpa* and *P. pseudostrobus* wood. In addition, as mentioned above, the impossibility
to find MWCNTs into the *P. montezumae* samples and the sole presence of amorphous carbon structures in this species, indicates that *P. montezumae* trees lacks either the concentration of catalystics or the type of metal required for an effective synthesis of these structures by a CVD like method (*Lara-Romero et al., 2011*; *Zhao et al., 2011*).

The presence of iron in the samples of *P. oocarpa* containing MWCNTs suggests that this metal provided catalytically active sites for the CNT synthesis. Previous studies that used plant tissue precursors to catalyze CNT growth have suggested that iron catalytic sites are uniformly distributed in plant cells (*Zhao et al., 2011*). Consequently, CNTs formed in plant tissues could be expected to have uniform diameters. This is consistent with our HR-TEM observations (average wall thickness ∼7–13 layers; inner diameter of ∼2–6 nm; outer diameter of ∼12–20 nm; length of 2.5–20 μm), which revealed that the MWCNTs had homogeneous number of layers and external diameters. In addition, the TGA results indicated that the burned wood samples of *P. oocarpa* after pyrolysis degradation contain ∼2.8% wt of CNTs. Although it has been generally accepted that nature could provide the conditions for their synthesis, there is scarce evidence of naturally formed MWCNTs in the biosphere. Therefore, we provide evidence of spontaneously and naturally produced MWCNTs from *Pinus* species, following forest wildfires.

In another context, the effects of nanomaterials such as CNTs on plant growth and development has been documented, and it has been suggested that their effects are because of factors such as the type of nanoparticles, concentration, plant species, and experimental conditions, including the method of nanoparticle uptake (*Tiwari et al., 2014*); in contrast, studies indicate that some CNTs nanomaterials show toxic effects on several plant models (*Miralles, Church & Harris, 2012*). With respect to human engineered MWCNTs, several molecular mechanisms involved in their biological effects have been described. Genomic analyses of *Lycopersicon esculentum* have indicated that exposure to MWCNTs altered the total gene expression, with up-regulation of stress-related genes (*Lahiani et al., 2015*; *Lahiani et al., 2016*), while that in *Nicotiana tabacum* has been found to cause alterations in total gene expression, with up-regulation of genes related to cell-wall assembly/cell growth, regulation of cell cycle progression, and aquaporin production (*Lahiani et al., 2015*; *Miralles, Church & Harris, 2012*; *Mukherjee et al., 2016*; *Yang, Cao & Rui, 2017*). Thus, the authors have suggested that size, composition, and specific surface characteristics of the engineered nanomaterials may play important roles in their phytotoxicity (*Hong, Peralta-Videa & Gardea-Torresdey, 2013*; *Mukherjee et al., 2016*).

In our work, the effect of MWCNTs was evaluated using two plant species found in the burned forest ecosystem, although the MWCNTs utilized are of synthetic origin; these were acquired with structural characteristics similar to those of CNTs found in burned wood samples from the resinous forest. Interestingly, seed germination and growth promotion were observed in both the *L. elegans* and *E. polystachya* plant species tested, with the influence of all the quantified biometrical plant variables; it was unlikely that seed germination and growth promotion occurred exclusively owing to water retention in the plant tissues (Fig. 6). In addition, the results did not indicate that CNTs had toxic effects on seed germination or plant development in the concentration range used, suggesting that at low dosages, MWCNTs function as plant-growth promoters. The plant

growth dose-dependence also suggests that the concentrations at which CNTs exert their maximum plant growth-promoting effect depend on the plant species. Although the mechanisms of the observed biological effects were not investigated, the findings indicate that the seed germination and plant-growth promotion was due to the activation of the cell division and nutrient uptake and also increased water influx as previously suggested, rather than an increase in the cell volume.

Forest fires are known to enhance the recruitment of a number of important native species associated with forest ecosystems (*Keeley & Fotheringham, 1998*; *Keeley et al., 2011*; *Turner et al., 1997*), including *E. polystachya* (*Orozco, 2008*) and *L. elegans* (*Díaz-Rodriguez et al., 2013*). In addition, products resulting from the combustion of wood, such as ash (*Keeley & Fotheringham, 1998*) and charred wood (*Roy & Sonie, 1992*), have also been shown to trigger germination and plant growth after forest fire events. The influence of MWCNTs formed in burned wood after forest wildfires on plant growth or other post-fire characteristic events in terrestrial ecosystems requires extensive studies.

CNT formation has usually been associated with extreme environments; however, we have provided evidence that MWCNTs can be found in biotic environments after atmospheric events. MWCNTs, formed in forest wildfires could be introduced into the soil by burned plant material such as smoke or solid particles. If this is true, then MWCNTs have been interacting with soil, organisms, and plants species since a long time. This may explain our findings, which strongly suggest that MWCNTs produced in resinous forest wildfires promote seed germination and growth of native plants in forest ecosystems.

## CONCLUSION

This study shows direct evidence of MWCNT generation during forest wildfires as a natural phenomenon, strongly suggesting a possible impact on natural plants of the resinous forest ecosystems through their effects on seed germination and plant growth promotion.

### Funding
This research was funded by CONACYT (256119) and C.I.C. 2.14/UMSNH grants. The funders had no role in study design, data collection and analysis, decision to publish, or preparation of the manuscript.

### Grant Disclosures
The following grant information was disclosed by the authors:
CONACYT: 256119.
C.I.C.: 2.14/UMSNH.

### Competing Interests
The authors declare there are no competing interests.
## Author Contributions

- Javier Lara-Romero performed the experiments, analyzed the data, contributed reagents/materials/analysis tools, wrote the paper, prepared figures and/or tables, reviewed drafts of the paper.
- Jesús Campos-García and Javier A. Villegas conceived and designed the experiments, analyzed the data, contributed reagents/materials/analysis tools, wrote the paper, prepared figures and/or tables, reviewed drafts of the paper.
- Nabanita Dasgupta-Schubert analyzed the data.
- Salomón Borjas-García, DK Tiwari and Roberto Lindig-Cisneros performed the experiments, analyzed the data, contributed reagents/materials/analysis tools.
- Francisco Paraguay-Delgado, Sergio Jiménez-Sandoval and Gabriel Alonso-Nuñez analyzed the data, contributed reagents/materials/analysis tools.
- Mariela Gómez-Romero performed the experiments, analyzed the data, reviewed drafts of the paper.
- Homero Reyes De la Cruz analyzed the data, reviewed drafts of the paper.

## Field Study Permissions

The following information was supplied relating to field study approvals (i.e., approving body and any reference numbers):

Sampling was collected under the supervision of the Ministry of Environment and Natural Resources specifications (Nom-059-SEMARNAT-2010) and the conservation program for flora and fauna of the Pico de Tancítaro (APFFPT) from Michoacán, México; established by the Mexican decree law of august 19, 2009; and the Program for the Sustainable Management of Mountain Ecosystems Pico de Tancítaro, Michoacán, México (APFFPT-2009).

## Data Availability

The raw data has been supplied as a Supplemental File.

## Supplemental Information

Supplemental information for this article can be found online at http://dx.doi.org/10.7717/peerj.3658#supplemental-information.

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
