# Peer review of "Biological effects of carbon nanotubes generated in forest wildfire ecosystems rich in resinous trees on native plants"

_PeerJ, doi:10.7717/peerj.3658_

## Round 0.1 · original submission · Minor Revisions

As you see, all three reviewer are quite positive. However, they have some minor remarks that you might want to address in a revised version.

Reviewer 1 ·

Basic reporting

good.
one paper cited has been retracted.

Experimental design

good

Validity of the findings

Overall good.
Repetition. It is common practice in plant and crop studies in order to be confident of the results to fully repeat experiments, each with replication. Were the germanation/growth experiments repeated?

Additional comments

General comments:

In this study, the authors showed that multiwalled carbon nanotubes (MWCNTs) can be formed in nature in forest fire without human interference, especially for wood samples of Pinus oocarpa and Pinus pseudostrobus. Also, the authors found that the seed germination rate and growth rate of Lupinus elegans and Eysenhardtia polystachya can be promoted by the presence of high concentrations of MWCNTs. The findings are actually quite exciting and the manuscript is very well written. The methods appear sound and the conclusions follow from the observations.
To strengthen the paper even further the authors need to provide a little more solid evidence for some of their conclusions, such as “P. oocarpa contained ~2.8% (w/w) of CNTs” and “MWCNT had same number of layers and external diameters”. And correct errors in a Figure. The manuscript will be acceptable if the following specific comments are adequately addressed.


Specific comments:


1) Line 154. The characteristic bands of the authentic CNTs should be obtained and shown for comparison with those of P. oocarpa and P. pseudostrobus.

2) Line 176. The conclusion “P. oocarpa contained 2.8% of CNTs” needs to be verified by observing the remaining substance right before increasing temperature to 610 °C via TEM. If a considerable fraction of substance are shown as CNTs, then the conclusion can be verified.

3) Figure 5. There are only a and b but no c in Figure 5. I think the authors mislabeled Figure 5b and 5c. Also, there is no label for y-axis in the current Figure 5b.

4) Line 262-263. Statistical analysis on diameters and wall layers of all the MWCNTs observed under TEM needs to be conducted in order to support the statement “MWCNTs had same number of layers and external diameters.”

5) I think the D and G bands would appear in any char formed at high temperature resulting from the graphitic microcrystallites that are produced. Their appearance does not necessarily signify the presence of CNTs. Please discuss.

6) Line 294-298. This is speculative and the paper that is cited (Khodakovskaya, 2009) has been retracted. It is best to leave out of the manuscript for that reason, and also because there is no evidence presented here that supports any particular mechanism.

Reviewer 2 ·

Basic reporting

no comment

Experimental design

no comment

Validity of the findings

no comment

Additional comments

This article describes direct evidence of MWCNT generation during forest wildfires as a natural phenomenon, suggesting a possible impact on natural plants of the resinous forest ecosystems through their effects on seed germination and plant growth promotion. However, in my opinion, the manuscript requires revisions before accepting.

1. It is well known that CNTs are hydrophobic, and the usual method is ultrasonic vibration. The authors should provide details about dissolution of synthetic MWCNTs (line 128-130).
2. The concentration of MWCNTs in the soil after forest wildfires should be measured in order to compare with the concentration of CNTs that affected the growth or germination of plant under the experimental condition.
3. The kind of software and statistical analysis that the authors utilized in this manuscript should be added in the manuscript.
4. The “Conclusion” section should be added into the manuscript.

Reviewer 3 ·

Basic reporting

This article adresses a very interesting subject and meets all the standards of PeerJ concerning the language, the context provided, structure and data presentation. I have no further comments.

Experimental design

Even though the introduction contextualizes well the research questions, in my opinion the objectives of the study are not clearly defined. I suggest the last paragragh of the Introduction section (page 8, lines 88-91) to be rephrased so that the objective of the work becomes compatible with the research questions described in the discussion (page 15, lines 246-47 and page 17, lines 284-85).

The experiments seem to be well designed and properly conducted. However, the authors should mention the number of wood samples collected from each burnt area. In addition, the number of samples characterised by Raman spectroscopy, thermogravimetry , and high-resolution transmission electron microscopy should also be mentioned.

Validity of the findings

Unexpectedly, no evidence of CNT structures was found in P. montezumae samples. The authors are encouraged to advance some hypotheses to explain this observation.

The conclusions are supported by the experimental data and are appropriately stated.

Additional comments

This article is well written, adresses a very interesting subject and meets all the standards concerning the scientific language, the context provided, structure, data analysis, presentation and discussion.

---

## Round 0.2 · accepted · Accept

You have addressed all the concerns of the three reviewers adequately. It is a very nice manuscript and I am looking forward to seeing it published.